# Analysis of the determinants for using health research evidence in health planning in Tanzania: a cross-sectional study

Pius Kagoma[1,2]*, Richard Mongi[1], Albino Kalolo[3,4]

1 Department of Public Health and Community Nursing, The University of Dodoma, Dodoma, Tanzania, 2 Division of Health, Social Welfare and Nutrition Services, President's Office-Regional Administration Local Government, Dodoma, Tanzania, 3 Department of Public Health, St. Francis University College of Health and Allied Sciences, Morogoro, Tanzania, 4 Centre for Reforms, Innovation, Health Policies, and Implementation Research, Dodoma, Tanzania

* piuskagoma@gmail.com

## Abstract

### Introduction

The use of health research evidence is essential for informed decision-making and effective health planning. Despite its importance, there is limited understanding of the determinants for the use of such evidence in planning processes, particularly in lower-middle-income countries (LMICs) like Tanzania. This study aims to investigate the proportion and determinants that affect the use of health research evidence in health planning in Tanzania.

### Materials and methods

This quantitative study employed a cross-sectional design. Data on health research evidence and the factors influencing its use were collected using a structured questionnaire from 422 healthcare workers involved in planning within 9 regions of Tanzania from October to December 2023. The association between categorical variables was assessed using a chi-square test, while regression analysis was conducted to identify determinants, both at a 95% confidence level,

### Results

The study revealed that 270 (66.2%) of health planning team members strongly agreed that they use health research evidence during planning. Several key determinants were significantly associated with the level of research evidence utilization. These included limited dissemination of research findings (74.5%), inadequate human and non-human resources (70.0%), and insufficient knowledge and training in research (63.7%). A multivariate regression analysis confirmed significant associations between the determinants and the use of research evidence (p<0.05).

**Data availability statement:** All relevant data are within the manuscript and its Supporting Information files.

**Funding:** The author(s) received no specific funding for this work.

**Competing interests:** The authors have declared that no competing interests exist.

**Abbreviations:** CCHP: Comprehensive Council Health Planning; CHOP: Comprehensive Hospital Operational Plan; LGAs: Local Government Authorities; LMICs: Lower-and Middle-Income Countries; MoH: Ministry of Health; PO-RALG: President's Office-Regional Administration and Local Government; RHMT: Regional Health Management Team; SDGs: Sustainable Development Goals; UHC: Universal Health Coverage; WHO: World Health Organization.

Descriptive statistics revealed that over 70% of respondents identified the presence of research coordinators, partnerships with universities, availability of research budgets, and internet access as important factors in their research. Inferential analysis indicated that these factors were statistically significantly associated with the use of health research evidence. In addition, more than half of the participants stated motivational factors, such as the presence of continuous quality improvement initiatives, the availability of short- and long-term training programs, on-the-job training opportunities, and incentives like extra duty allowances, as contributors to the enhanced use of research evidence.
**Bottom of Form**

## Conclusion

The study found that planning team members used health research evidence in planning, but several determinants, such as lack of dissemination, resource shortages, and inadequate training, persisted. Interventions should focus on improving dissemination, resources, and training. Future research should explore strategies for enhancing these interventions.

## Introduction

Health research evidence plays an essential role in shaping effective health policies and interventions, which are crucial for improving public health outcomes [1]. In resource-limited settings like Tanzania, where health challenges are complex and resources are constrained, the strategic use of research evidence is particularly important. It can guide decision-makers in identifying priority areas, optimizing resource allocation, and implementing interventions that are both cost-effective and impactful. Moreover, the use of evidence can help health planning teams avoid repeating past failures and introduce innovative solutions [2]. Despite its recognized importance, the extent to which health research evidence is utilized by health planning teams in Tanzania remains unclear. Understanding the factors that influence this utilization is essential for promoting evidence-based decision-making, which is critical for achieving the ambitious health targets set forth in the Sustainable Development Goals (SDGs), particularly Universal Health Coverage (UHC) [3].

Efforts to promote the integration of research evidence into health planning and decision-making have been made in various settings. These include the establishment of knowledge translation platforms, research dissemination strategies, health conferences, and capacity-building initiatives targeting health planning teams. However, the success of these efforts varies significantly across contexts, largely depending on factors such as institutional support, the capacity of the health workforce, and access to reliable data. In Tanzania, while some initiatives aim to bridge the gap between research and practice, the actual impact of these efforts on health planning remains under-explored.

Research across different settings has demonstrated that the use of health research evidence in decision-making is influenced by a variety of factors. These include organizational culture, the availability of resources, the expertise of the planning teams, and access to research outputs. In a study conducted in Tanzania [4], it was found that health planning teams in Tanzania consist of professionals from various disciplines, including Administrators (9%), Medical doctors (25.36%), Nurses (23.46%), Laboratory scientists (8.53%), Pharmacists (6.64%), Radiographers (1.66%), Environmental health officers (3.08%), Nutrition officers (3.08%), Social welfare officers (4.74%), and others (14.46%). Their primary responsibility is to plan, implement, and evaluate health interventions [4]. However, several factors may limit their ability to integrate research evidence into their work fully [5]. These include inadequate dissemination of research findings, lack of access to research evidence, limited skills in interpreting research data, and insufficient organizational support [3,6–8].

While previous studies have explored various factors influencing the use of health research evidence in policy and planning, there remains a gap in understanding the specific determinants that affect the utilization of health research evidence among health planning teams in Tanzania [4,9]. This study seeks to fill this gap by analyzing the key factors that influence evidence use among these teams at both the regional and council levels, guided by the COM-B Model (Capability, Opportunity, and Motivation) as the theoretical framework [10], which has been widely used in understanding evidence-based decision-making processes. The COM-B model is particularly suited to the Tanzanian context because it focuses on the essential components of behavior change: Capability, Opportunity, and Motivation, which are critical for understanding and addressing the factors influencing the use of health research evidence by planning teams. By examining these elements, the model helps identify specific determinants that can be targeted to improve evidence-based decision-making in Tanzania's resource-limited setting. By applying this theoretical lens, the study aims to provide a deeper understanding of the determinants of using research evidence in health planning, with the following specific objectives: 1) Analyze the current usage of health research evidence among planning team members at the regional and council levels, 2) Analyze the capacity of health planning team members to utilize health research evidence, 3) Identify opportunities for enhancing the use of health research evidence in health planning, 4) Identify opportunities for enhancing the use of health research evidence in health planning, 5) Explore the motivations for using health research evidence among health planning members at both regional and council levels. The findings of this study are expected to contribute to the development of strategies that enhance the integration of research evidence into health planning processes, ultimately leading to more effective health interventions and improved health outcomes in Tanzania.

### Conceptual framework

The conceptual framework for this study was developed after several consultative meetings with different stakeholders and researchers. The theory was adopted and modified from the (Capability, Opportunity, and Motivation) COM-B model. The COM-B Model at the center of a proposed framework is a behavior system involving three essential conditions: Capability, Opportunity, and Motivation, which we term the 'COM-B system [10,11]. This study investigated how the capability, opportunity, and motivation determinants can influence the behavior of the use of health research evidence during health planning, as shown in Fig 1

## Materials and methods

### Study setting

This study was conducted between October and December 2023 in the United Republic of Tanzania, a Lower and middle-income country located in East Africa, with a population of about 62 million people. The allocated budget for the country's Ministry of Health in 2023/2024 was estimated to be 443.6 million US dollars, with 1.2 million US dollars (0.27%) allocated for evidence production. The health research evidence users in Tanzania involve three important Ministries, namely the Ministry of Health (MoH), the President's Office-Regional Administration and Local Government (PO-RALG), and the Health Management. The health evidence producers in Tanzania are the National Institute for Medical Research (NIMR), the Tanzania

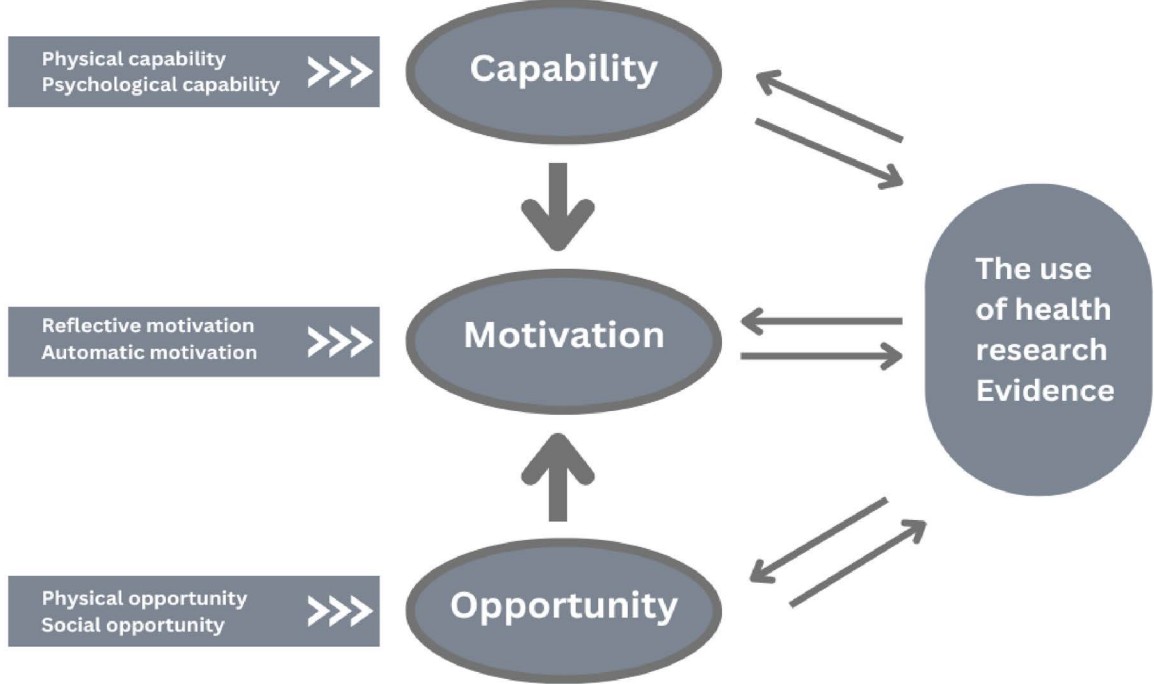

**Fig 1. The domains of the COM-B Model modified from Michie et al., 2011 [ 10].**

Commission for Science and Technology (COSTECH), Public and Private Universities, health-related institutions or authorities, local and international non-governmental organizations (NGOs), and civil society organizations, as shown in Table 1.

## The study sites

The study site was eighteen [18] Councils of the Local Government Authorities within the nine [9] regions out of 26 Regions of Tanzania Mainland from nine [9] geographical zones as summarized in Table 2. The nine zones are selected to seek the country's geographical representation. Together, these regions have a total population of 21,119,700, which represents 35.7% of the Tanzanian population. These regions are heterogeneous in population size, distribution of health facilities, human resources for health, and institutions carrying out health research activities. A similar approach has been used in major Tanzanian health studies [4,5,12]; This approach provided a comprehensive understanding of the status quo for the use of health research evidence in health planning, and a heterogeneous study population for random sampling, hence acting as a representative snapshot of all regions in Tanzania. The study was conducted in 63 randomly selected health facilities, nine [9] Regional Referral Hospitals (RRHs), eighteen [18] District Hospitals (DHs), eighteen [18] Health Centres (HCs), and eighteen [18] Dispensaries. To ensure the inclusion of urban and rural health facilities, stratification was conducted, followed by random sampling.

## Study population

This study involved members of health planning teams at the regional and council levels in Tanzania. Eligible participants were those holding health planning roles in healthcare facilities or administrative positions, with at least one year of experience in health planning. Individuals who were not part of the health planning team or had less than one year of experience were excluded from the study.

**Table 1. A list of health research evidence producers and users in Tanzania.**

| SN | Evidence users |
|---|---|
| 1 | Ministry of Health |
| 2 | President's Office-Regional Administration and Local Government |
| 3 | Ministry of Education, Science and Technology (MEST) |
| 4 | Prime Minister's Office |
| 5 | Regional Health Management Teams |
| 6 | Council Health Management Teams |
| 7 | Healthcare workers |
| 8 | Non-Governmental Organizations (NGOs) |
| 9 | Civil Society Organizations (CSOs) |
| 10 | Private Sector |
| **Evidence producers** | |
| 1 | National Institute for Medical Research (NIMR) |
| 2 | Tanzania Commission for Science and Technology (COSTECH) |
| 3 | Public Universities: The Muhimbili University of Health and Allied Sciences (MUHAS), University of Dar es Salaam (UDSM), University of Dodoma (UDOM) |
| 4 | Private Universities: Catholic University of Health and Allied Sciences (CUHAS), Kilimanjaro Christian Medical Centre (KCMC), Hubert Kairuki Memorial University (HKMU), St. John's University of Tanzania (SJUT), St. Augustine University of Tanzania (SAUT), Kampala International University (KIU), St. Francis University College of Health and Allied Sciences (SFUCHAS) |
| 5 | Kilimanjaro Clinical Research Institute (KCRI) |
| 6 | Ifakara Health Institute (IHI) |
| 7 | Non-Governmental Organizations (NGOs) |
| 8 | Civil society organizations (CSOs) |
| 9 | Private sector |

## The health planning landscape in Tanzania

The health planning process in Tanzania is conducted at two levels: At the Council level, the health facility prepares its plans as well, and the Council Health Management Team (CHMT) prepares its plan, whereby plans from the facility and CHMT are later consolidated to form the Comprehensive Council plan (CCHP). The preparation of health facility plans is guided by the Health Facility Planning Guidelines and CCHP guidelines. [13]. At the regional level, the Regional Health Management Teams (RHMTs) prepare their plans using the RHMT planning guide, and the Regional Referral Hospitals (RRHs) plans using the Comprehensive Operational Plan Guide (CHOP) [14]. When the plans are completed are sent to the Ministry of Health and the President's Office-Regional Administration and Local Government for final assessment before being sent to the Ministry of Finance for funding. At all levels, we are planning to use the guidelines and the routine data collected from each level [14].

## Study design

The study employs a quantitative approach with a cross-sectional design focusing on nine regions of Tanzania. [15,16].

## Sample size and sampling procedures

The sample size involved the health planning team members from the RRHs, DHs, HCs, and Dispensaries randomly selected from the nine regions. The sample size was 422, calculated from the Cochrane formula (1977). To date, in Tanzania, there is no cited reference for the percentage of the use of health research evidence in health planning; therefore, the assumed proportion of 50% was used [17].

**Table 2. Summary of study sites.**

| ZONE | REGION | COUNCIL |
|---|---|---|
| Northern | | Siha District |
| | Kilimanjaro | Moshi Municipal |
| Central zone | | Chamwino District |
| | Dodoma | Dodoma City |
| Dar es Salaam | | Kigamboni Municipal |
| | Dar es Salaam | Dar es Salaam City |
| Eastern | | Morogoro Municipal |
| | Mororgoro | Gairo District |
| Western | | Kigoma District |
| | Kigoma | Kigoma Municipal |
| Southwest highland | | Mbeya City |
| | Mbeya | Kyela District |
| Lake | | Kwimba District |
| | Mwanza | Mwanza City |
| Southern | | Mtwara Municipal |
| | Mtwara | Nanyamba |
| Southern Highland | Iringa | Iringa Municipal |
| | | Iringa District |

$$n = \frac{Z^2 * p\,(1-p)}{d^2}\;(\text{Cochran}\,(1977))$$

Where by

 N-sample size

Z-confidence interval 95% 1.96, P-proportional from previous study, d = margin of error, which is approximately 5%. Thus,

$$n = \frac{(1.96)^2 \text{ x } 50\,(100 - -50)}{(5)^2}$$

**=384** including 10% of non-response sample 384 plus 10% = 38 + 384 = 422

This study employed a multistage sampling technique for the selection of the study units, as summarized in Fig 2. The sampling stages are zones, regions, councils, and public primary health facilities. The first stage was a random selection of one region from each of the nine Zones of the country. In the second stage, in each selected region, councils were clustered into rural and urban, and then one rural and one urban council were selected from each region, followed by a random selection of the health facilities (see Fig 3. The technique is convenient for studying large and diverse populations [5]. The sampling stages were zones, regions, councils, and public primary health facilities.

## Measurement of variables

**Dependent variable.** The dependent variable of this study, the use of health research evidence, was measured using a set of four [4] questions adapted from a previous study [18]. The questions were a mixture of binary and multiple-choice questions. Those questions were divided into four areas. The mean score was calculated from those sets of questions;

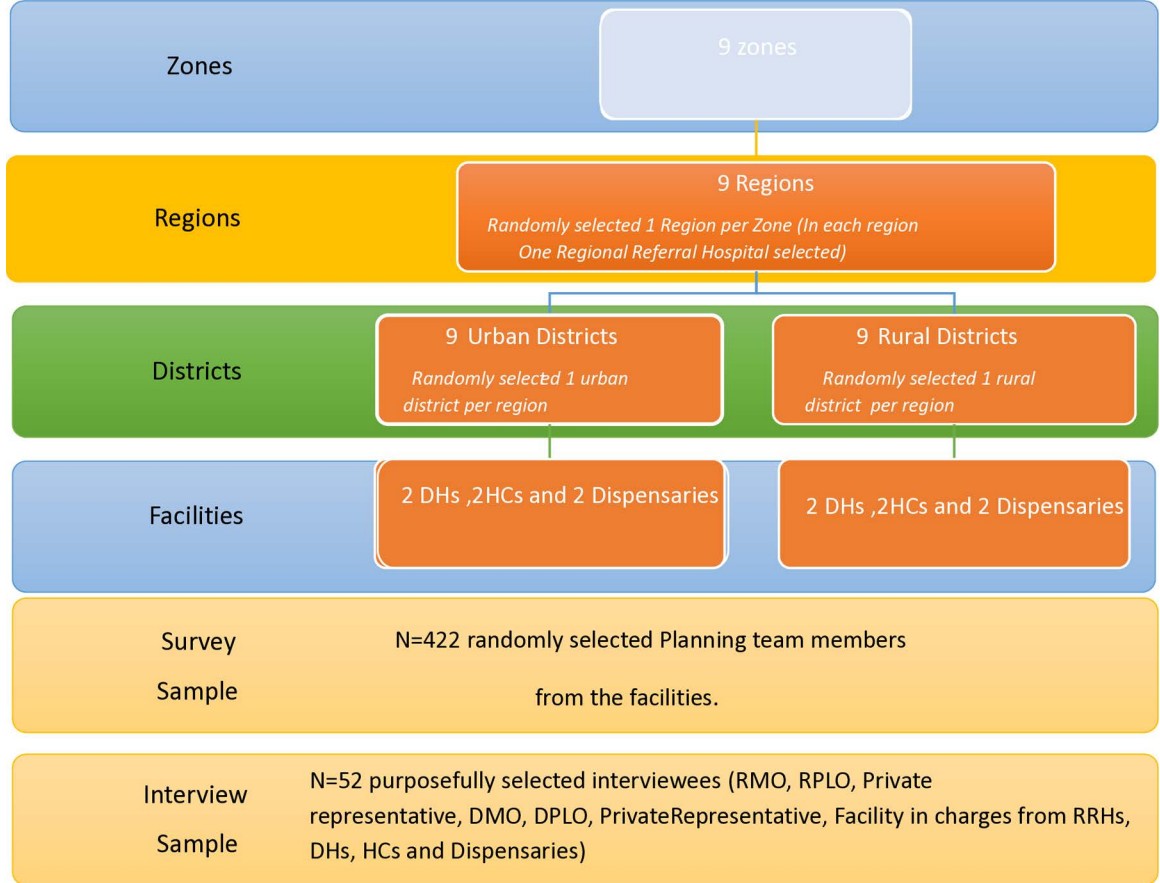

**Fig 2.** Multistage sampling technique for the selection of the study units.

the score had a value of 0 or 1. Those respondents who scored 1 were regarded as having used the health research evidence, while those who scored 0 had not used the health research evidence. For multiple choice questions, the respondents were provided with a set of predefined options, allowing them to select the one that best represents their response, which will capture varying degrees or categories on the use of health research evidence.

**Independent variables.** The independent variables in this study encompass numerous factors, including demographic information such as sex, age, and education level; professional attributes such as position within the health facility and years of schooling; and contextual elements including the type of health facility, stakeholders group represented, district, and region. The study aimed to gain insights into the determinants that influence the use of health research evidence during health planning. The determinants had a combination of three constructs, which are capability determinants, opportunity determinants, and motivation determinants, all derived from the COM-B Model.

**Capability determinants** were assessed using 12 Likert-scale questions [19] (1 = strongly agree to 5 = strongly disagree) and one multiple-choice question. Mean scores and standard deviations were calculated, and chi-square tests were used for categorical responses.

**Opportunity determinants** were evaluated through 17 Likert-scale questions, with responses assigned numerical values [1–5]. Mean or median scores were used to quantify overall tendencies, and multiple regression analysis was conducted to determine their influence on the dependent variable.

**Motivation determinants were** measured using 8 Likert-scale questions, analyzed similarly to opportunity determinants, with multiple regression analysis applied to assess relationships with the dependent variable.

**Data collection procedures and tools.** Quantitative data was collected face to face by using the Swahili version of the survey after translation of the English version of the survey and document review checklist (see supplemental file) to planning team members at national, regional, council, and health facility levels. The questions from the tool were adopted and modified from previous studies [18,20]. The survey collected information on the use of health research evidence and its determinants from the sampled participants using Open Data Kit (ODK).

The document review checklist was used to guide the document review to see whether the available plans have any evidence of being prepared using health research evidence. The checklist contains a list of questions that will help review the plans made at the facility, council, regional, and national levels. The checklist is used to collect information on the available health plans, whether they are prepared using health research evidence. Data was collected by a research assistant, who was trained for three days in data collection methods, tools, and ethics. They were selected according to their experience in the health system. Pilot testing of the tools was conducted at Dodoma RHMT and Chamwino District Council at Chamwino CHMT, Chamwino Council Hospital, Chamwino Health Centre, and Buigiri Dispensary to enhance the validity, reliability, and effectiveness of the tool in capturing the intended information.

## Data analysis

Responses were reported on all survey items in all response categories, and summarized using frequencies and percentages for categorical variables, and means and standard deviation for non-categorical variables were computed. Given

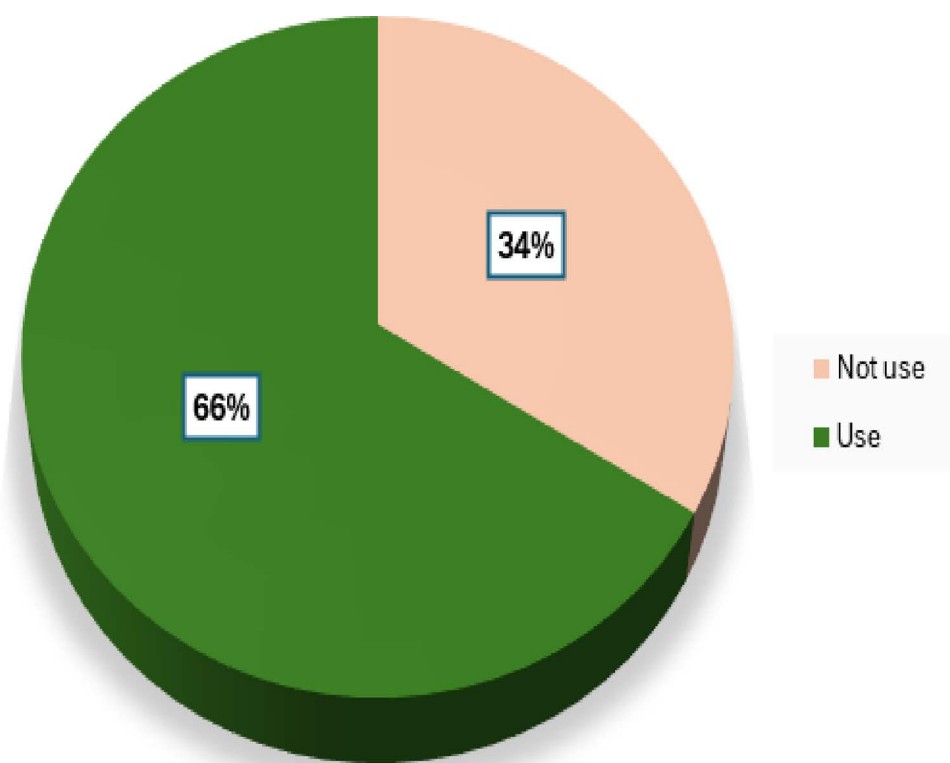

**Fig 3. Proportion of the use of health research evidence.**

that the outcome variable had two categories (0 = Negative, 1 = Positive), A binary logistic regression model was used to assess factors associated with the perception of implementers on the prime vendor system. The model results are regression parameter estimates and odds ratios (OR). The data analysis was conducted using STATA version 18, and the significance of all statistical tests was established at a 5% significance level.

## Data validity and reliability

To ensure validity and reliability, the validity of quantitative data collection tools was reviewed by independent subject matter experts from the University of Dodoma (UDOM). The reliability assessment of the questionnaire was done using the internal consistency test, with the alpha reliability coefficient being the statistic. [21,22]. The range of the alpha coefficient, also known as Cronbach's alpha, was 0.71 acceptable for this study. Moreover, an exploratory factor analysis was conducted to establish the construct validity of the questionnaire, and a pilot conducted at Chamwino DC before field data collection to ensure clarity of the data collection tool.

## Ethics and dissemination

The study was granted ethical approval by the University of Dodoma Ethical Committee with **Ref. No. MA.84/261/102/'A'/64/91.** Permission to conduct the study and consent to participate in the study were sought from relevant authorities and participants, respectively. Participants received information about the purpose of the study and data protection. Permission to conduct the study was also obtained from the Local Government authorities and the Ministry of Health. Written informed consent was obtained from all participants before their involvement in the study. Participants were informed of their right to withdraw from the study at any time, and their participation was entirely voluntary. Confidentiality was strictly maintained by ensuring the anonymity of all documents containing participants' information.

## Results

### Demographic characteristics

Table 1 describes the respondents' characteristics and the stakeholder group they represent. There was an almost equal number of female 50.2% and male 49.8% respondents, with a mean age of 39.51 ± 7.86years old. Most of them (42.7%) had an undergraduate education, followed by a Diploma (36.3%), and less than one-third (14.0%) had a master's degree and above. Few (7.1%) respondents had certificate-level education. Professional background results revealed that the majority, 107 (25.4%) of the respondents were medical doctors, followed by nurses, 99 (23.5%). Physiotherapists and planning offices were very few, each accounting for 0.24% of the respondents (Table 3).

### Proportion of the use of health research evidence

Table 4 provides data on the use of health research evidence in a particular context. It shows that the majority of participants (96.7%) have used routine data, with 49.5% reporting a high extent of use. Additionally, evidence from routine data was the most commonly used type of evidence during health planning, with 98.8% of participants reporting its use. Most participants (66.2%) have used health research evidence, as shown in Fig 3, and a significant majority (98.8%) have used health planning guidelines. Moreover, the majority of participants (98.8%) also indicated the importance of the use of health research evidence, with 82.6% considering it very important. These findings suggest a strong reliance on routine data and a high recognition of the importance of health research evidence in the context studied.

### The factors associated with the use of health research evidence during health planning

**Region**: The analysis compares various regions to the reference category, Kilimanjaro. The odds ratios (OR) show how likely individuals from other regions are to use health research evidence in health planning compared to those from Kilimanjaro.

**Table 3. Demographic characteristics of respondents.**

| Variable | Frequency | Percent |
|---|---|---|
| **Age** | | |
| <25 | 3 | 0.7 |
| 26-35 | 159 | 37.7 |
| 36-45 | 156 | 37.0 |
| 46-55 | 93 | 22.4 |
| >46 | 11 | 2.6 |
| **Region** | | |
| Mbeya | 48 | 11.4 |
| Dodoma | 50 | 11.9 |
| Iringa | 41 | 9.7 |
| Mtwara | 49 | 11.6 |
| Mwanza | 50 | 11.85 |
| Kigoma | 51 | 12.1 |
| Kilimanjaro | 45 | 10.7 |
| Morogoro | 48 | 11.4 |
| Dar es salaam | 40 | 9.5 |
| **Type of health facility** | | |
| Dispensary | 37 | 8.8 |
| Health center | 82 | 19.4 |
| Council Hospital | 97 | 23.0 |
| Regional referral hospital | 81 | 19.2 |
| RHMT | 87 | 20.6 |
| CHMT | 38 | 9.0 |
| Gender | | |
| Male | 210 | 49.76 |
| Female | 212 | 50.24 |
| **Professional background** | | |
| Doctor | 107 | 25.36 |
| Nurse | 99 | 23.46 |
| Laboratory scientist | 36 | 8.53 |
| Pharmacist | 28 | 6.64 |
| Radiographer | 7 | 1.66 |
| Environmental Health officer | 13 | 3.08 |
| Nutrition officer | 13 | 3.08 |
| Social Welfare Officer | 20 | 4.74 |
| Physiotherapist | 1 | 0.24 |
| Biomedical engineer | 3 | 0.71 |
| Planning officer | 1 | 0.24 |
| Health Secretary | 38 | 9.00 |
| Other | 56 | 13.3 |
| **Stakeholders' group** | | |
| RHMT | 49 | 11.6 |
| HMT | 280 | 66.4 |
| CHMT | 90 | 21.3 |
| FBO | 1 | 0.2 |
| HGFC | 2 | 0.5 |

*(Continued)*

**Table 3.** (Continued)

| Variable | Frequency | Percent |
|---|---|---|
| **Highest level of education** | | |
| Certificate | 30 | 7.1 |
| Diploma Certificate | 153 | 36.3 |
| Undergraduate degree | 180 | 42.7 |
| Master's Degree | 56 | 13.3 |
| PhD | 3 | 0.7 |
| **The position holds currently in health planning.** | | |
| Chairperson | 30 | 7.1 |
| Secretary | 45 | 10.7 |
| Technical Advisor | 11 | 2.6 |
| Member | 335 | 79.4 |
| Other | 1 | 0.2 |
| **Years participate** | | |
| <5 | 283 | 67.1 |
| 6-10 | 103 | 24.4 |
| >10 | 36 | 8.5 |

For instance, if a region has an OR greater than 1, individuals from that region are more likely to use research evidence than those from Kilimanjaro. If the OR is less than 1, they are less likely. The p-values indicate whether these differences are statistically significant.

**Level of Education**: Individuals with a diploma (reference category) than those with higher education levels were more likely to use research evidence:

Undergraduate degree (OR = 2.396, CI: [1.094, 5.249], p-value = 0.0290): Those with an undergraduate degree are over twice as likely to use research evidence than diploma holders.

Master's degree (OR = 2.278, CI: [0.921, 5.631], p-value = 0.0747): Similar to undergraduates, master's degree holders are also more likely to use research evidence in health planning, although these were not significant.

These OR values reflect the increased likelihood of individuals with higher education levels using research evidence compared to the reference group, with p-values indicating the strength of this relationship.

**Years of participation.** 6–10 years (OR = 1.092, CI: [0.677,1.761], p-value = 0.7184): Individuals with 6–10 years of participation in health planning show a higher likelihood of using research evidence than those with fewer years, though this was not statistically significant.

More than 10 years (OR = 1.399, CI: [0.648,3.019], p-value = 0.3924): Those with over 10 years of participation have a 40% higher likelihood of using research evidence than those with fewer years of experience, and this difference was also not statistically significant.

The adjusted odds ratios reflect the likelihood of each category of the variable influencing the use of research evidence, controlling for other variables in the model. Significance is determined by both the size of the OR and the p-value, with confidence intervals providing a range within which the true effect likely falls (Table 5).

Table 6 presents the findings from a survey on the capacity to use health research evidence (knowledge and skills). Each row in the table represents a different variable related to the use of health research evidence, and the columns display the responses categorized into five levels of importance: Very Unimportant (VUIM), Unimportant (UIM), Neutral, Important (IM), and Very Important (VIM).

**Table 4. The proportion of the use of health research evidence during health planning.**

| Variable | Frequency | Percent |
|---|---|---|
| **Ever used routine data** | | |
| No | 14 | 3.3 |
| Yes | 408 | 96.7 |
| **Extent of use of routine data** | | |
| Low | 41 | 10.1 |
| Medium | 165 | 40.4 |
| High | 202 | 49.5 |
| **Type of evidence used during health planning (Multiple response)** | | |
| Evidence from routine data | 403 | 98.8 |
| Policy documents | 280 | 68.6 |
| Research publications (general) | 133 | 32.6 |
| Systematic reviews/ meta-analysis | 41 | 10.1 |
| Randomized Control trials | 17 | 4.2 |
| Experimental studies | 30 | 7.4 |
| Non-experimental studies | 20 | 4.9 |
| Expert opinions | 271 | 66.4 |
| Policy beliefs | 162 | 39.7 |
| Other | 17 | 4.2 |
| **Use of health research evidence** | | |
| No | 138 | 33.8 |
| Yes | 270 | 66.2 |
| **Use of the Health planning guidelines** | | |
| No | 5 | 1.2 |
| Yes | 403 | 98.8 |
| **Use of the Ruling party manifesto** | | |
| No | 44 | 10.8 |
| Yes | 364 | 89.2 |
| **Use of the Policy documents** | | |
| No | 137 | 33.6 |
| Yes | 271 | 66.4 |
| **Importance of the use of health research evidence** | | |
| No | 5 | 1.2 |
| Yes | 403 | 98.8 |
| **Level of importance of the use of health research evidence** | | |
| Very unimportant | 5 | 1.2 |
| Unimportant | 2 | 0.5 |
| Neutral | 14 | 3.4 |
| Important | 50 | 12.3 |
| Very important | 337 | 82.6 |

From the data, it's clear that the majority of respondents perceive the use of health research evidence as very important or important across the various variables. In the variable "Gives the latest information," the majority of respondents rated it as Very Important (65.7%) or Important (24.5%). Similarly, in the variable "Help the policymakers understand a specific problem," a substantial proportion of respondents indicated that it is Very Important (66.7%) or Important (24.3%).

**Table 5. Binary logistic regression for factors associated with the use of research evidence during health planning.**

| Variable | Not use | Use | Adjusted logistic analysis | |
|---|---|---|---|---|
| | N (%) | N (%) | OR [95%CI] | p-value |
| **Age** | | | | |
| <35 | 52(32.1) | 110(67.9) | | |
| 36-45 | 56(35.9) | 100(64.1) | | |
| >45 | 35(33.7) | 69(66.4) | | |
| **Region** | | | | |
| Kilimanjaro | 26(57.8) | 19(42.2) | ref | |
| Mbeya | 17(35.4) | 31(64.6) | 2.604 [1.074,6.315] | 0.0342 |
| Dodoma | 17(34.0) | 33(66.0) | 2.514 [1.053,6.001] | 0.0378 |
| Iringa | 10(24.4) | 31(75.6) | 4.707 [1.772,12.499] | 0.0019 |
| Mtwara | 21(42.9) | 28(57.1) | 1.801 [0.727,4.466] | 0.2039 |
| Mwanza | 12(24.0) | 38(76.0) | 4.794 [1.893,12.140] | 0.0009 |
| Kigoma | 10(19.6) | 41(80.4) | 5.922 [2.224,15.769] | 0.0004 |
| Morogoro | 17(35.4) | 31(64.6) | 2.180 [0.901,5.274] | 0.0839 |
| Dar es salaam | 13(32.5) | 27(67.5) | 3.229 [1.211,8.611] | 0.0192 |
| **Type of health facility** | | | | |
| Dispensary | 17(46.0) | 20(54.1) | ref | |
| Health center | 25(30.5) | 57(69.5) | 2.029 [0.829,4.964] | 0.1211 |
| Council Hospital | 45(46.4) | 52(53.6) | 0.809 [0.323,2.026] | 0.6511 |
| RRH | 19(23.5) | 62(76.5) | 2.663 [1.895,7.920] | 0.0383 |
| RHMT | 27(31.0) | 60(69.0) | 0.350 [0.031,3.902] | 0.3932 |
| CHMT | 10(26.3) | 28(73.7) | 5.647 [0.602,52.94] | 0.3167 |
| Gender | | | | |
| Male | 71(33.8) | 139(66.2) | | |
| Female | 72(34.0) | 140(66.0) | | |
| **Professional background** | | | | |
| Doctor | 37(34.6) | 70(65.4) | | |
| Nurse | 35(35.4) | 64(64.7) | | |
| Lab scientist &Pharmacist | 25(35.2) | 46(64.8) | | |
| Non-medical | 28(31.5) | 61(68.5) | | |
| Other | 18(32.1) | 38(67.7) | | |
| **Stakeholders' group** | | | | |
| RHMT | 13(26.5) | 36(73.5) | 0.843 [0.177,4.012] | 0.8297 |
| CHMT | 26(28.9) | 64(71.1) | 5.647 [0.602,52.93] | 0.1295 |
| HMT | 104(36.8) | 179(63.3) | ref | |
| **Highest level of education** | | | | |
| Certificate | 15(50.0) | 15(50.0) | ref | |
| Diploma | 57(37.3) | 96(62.8) | 1.609 [0.670,3.867] | 0.2874 |
| Undergraduate | 53(29.4) | 127(70.6) | 1.824 [0.710,4.685] | 0.2120 |
| Master's Degree | 18(30.5) | 41(69.5) | 2.082 [0.684,6.337] | 0.1968 |
| **Position holds currently in health planning** | | | | |
| Secretary | 19(42.2) | 26(57.8) | ref | |
| Chairperson | 8(26.7) | 22(73.3) | 1.933 [0.647,5.780] | 0.2381 |
| Technical | 4(36.4) | 7(63.6) | 0.789 [0.173,3.599] | 0.7600 |
| Member | 112(33.3) | 224(66.7) | 1.328 [0.651,2.708] | 0.4354 |

*(Continued)*

| Variable | Not use | Use | Adjusted logistic analysis | |
|---|---|---|---|---|
| **Years participate** | | | | |
| <5 | 99(35.0) | 184(65.0) | | |
| 6-10 | 34(33.0) | 69(67.0) | | |
| >10 | 10(27.8) | 26(72.2) | | |

Capability: The knowledge and skills of health research evidence use.

**Table 6. Knowledge and skills of health research evidence use.**

| Variable | VUIM n (%) | UIM n (%) | Neutral n (%) | IM n (%) | VIM n (%) |
|---|---|---|---|---|---|
| Gives the latest information | 5(1.2) | 3(1.2) | 32(7.8) | 100(24.51) | 268(65.7) |
| Help the policymakers understand a specific problem | 3(0.7) | 4(1.0) | 30(7.4) | 99(24.3) | 272(66.7) |
| Help the policymakers to implement various health interventions | 3(0.7) | 5(1.2) | 24(5.9) | 113(27.7) | 263(64.5) |
| Avoid repeating the failures of others | 3(0.7) | 3(0.7) | 19(4.7) | 101(24.8) | 282(69.1) |
| Introduces health planners to new ideas | 3(0.7) | 2(0.5) | 19(4.7) | 107(26.2) | 277(67.9) |

**Table 7. Obstacles that hinder the utilization of health research evidence.**

| Variable | SDSA n (%) | DSA n (%) | Neutral n (%) | AG n (%) | SAGR n (%) |
|---|---|---|---|---|---|
| Lack of necessary knowledge and training in research | 12(2.94) | 18(4.4) | 7(1.7) | 67(16.4) | 304(74.5) |
| Inadequate human and non-human resources | 16(3.92) | 18(4.4) | 7(1.7) | 81(19.9) | 286(70.1) |
| Difficult in accessing health research evidence | 36(8.82) | 41(10.1) | 24(5.9) | 110(27.0) | 197(48.3) |
| Perceptions that health research is for academic purposes | 92(22.55) | 40(9.8) | 16(3.9) | 81(19.9) | 179(43.9) |
| Lack of involvement in research activities | 15(3.68) | 25(6.1) | 11(2.7) | 93(22.8) | 264(64.7) |
| Traditional ways of planning and Minimal usage of health research evidence among planning team members at the regional and council levels | 36(8.82) | 21(5.2) | 13(3.2) | 92(22.6) | 246(60.3) |
| lack of dissemination | 11(2.70) | 15(3.7) | 17(4.2) | 105(25.7) | 260(63.7) |

Overall, the data suggests that there is a strong recognition of the significance of health research evidence in various aspects of policymaking and intervention implementation.

Table 7 provides information on the obstacles hindering the utilization of health research evidence. The variables include Lack of necessary knowledge and training in research, Inadequate human and non-human resources, Difficulty in accessing health research evidence, Perceptions that health research is for academic purposes, Lack of involvement in research activities, and Traditional ways of planning, as well as the minimal usage of health research evidence among planning team members at the regional and council levels, and lack of dissemination.

The data is presented in terms of the number and percentage of participants who strongly disagree (SDSA), disagree (DSA), are neutral, agree (AG), and strongly agree (SAGR) with each obstacle. This data provides insight into the specific challenges that impact the utilization of health research evidence within the context of the study. The highest percentage is for "Perceptions that health research is for academic purposes" at 22.6%, and the lowest percentage is for "Lack of necessary knowledge and training in research" at 2.9%.

## Opportunities for the use of health research evidence

Table 8 indicates the opportunities for the use of health research evidence, helping stakeholders to identify areas for improvement and investment to optimize the use of health research evidence. The table presents the opportunities for utilizing health research evidence, categorized into physical and social opportunities.

For physical opportunities, the table highlights the availability of resources such as research coordinators, equipment, internet access, a planned budget for research, and the presence of symposiums for research discussions. The percentages indicate the distribution of responses, ranging from strongly disagree to strongly agree, reflecting the varying levels of agreement among the respondents regarding these physical opportunities. In the physical opportunities category, the highest percentage is 76.3%, representing the availability of the internet, while the lowest percentage is 0.7%, indicating the availability of Internet.

In the social opportunities section, the table emphasizes the presence of disparities among different populations, the involvement of community members in health planning teams, and the enhancement of health literacy in society. The percentages provide an insight into the attitudes and perceptions of the respondents toward these social opportunities. In the social opportunities category, the highest percentage is 55.9%, reflecting improved health literacy in society, and the lowest percentage is 2.4%, which represents the presence of disparities among different populations, allowing for more equitable representation.

## Motivations for the use of health research evidence

Table 9, based on the data presented, we can observe that the automatic motivations for the use of health research evidence are primarily centered around the availability of transparency and accountability mechanisms 66.4% strongly agreeing, and 23.2% agreeing with the statement availability of job training 73.2% strongly agreeing and 21.3% agreeing respectively. Provision of incentives to health planners, with 72.4% strongly agreeing and 15.4% agreeing respectively.

In contrast, reflective motivations such as continuous quality improvement in the healthcare system and the presence of active stakeholder engagement in health planning also garnered significant support, with 63.51% and 59.48% strongly agreeing or agreeing, respectively. This indicates a strong emphasis on the proactive involvement of various stakeholders and the continuous enhancement of the healthcare system based on research evidence.

## Findings from the document review checklist

The analyzed data from Table 10 indicates that the majority of planning teams incorporate research evidence into their health plans. A substantial 97.8% of respondents confirmed having a current fiscal year health plan, and 98.9% reported

**Table 8. Opportunities for the use of health research evidence.**

| Variable | SDSA n (%) | DSA n (%) | Neutral n (%) | AG n (%) | SAGR n (%) |
|---|---|---|---|---|---|
| **Physical opportunities** | | | | | |
| Enabling the utilization of information generated from the healthcare system | 6(1.4) | 9(2.1) | 18(4.3) | 130(30.8) | 259(61.4) |
| Availability of research coordinators | 9(2.1) | 18(4.3) | 22(5.2) | 93(22.0) | 280(66.4) |
| Availability of equipment | 4(1.0) | 16(3.8) | 21(5.0) | 88(20.9) | 293(69.4) |
| Availability of internet | 3(0.7) | 9(2.1) | 19(4.5) | 69(16.4) | 322(76.3) |
| Availability of planned budget for research | 11(2.6) | 17(4.0) | 27(6.4) | 64(15.2) | 303(71.8) |
| Presence of symposiums for research discussions | 9(2.1) | 17(4.0) | 23(5.5) | 106(25.1) | 267(63.3) |
| **Social opportunities** | | | | | |
| The presence of disparities among different populations allows for more equitable | 10(2.4) | 24(5.7) | 38(9.0) | 147(34.8) | 203(48.1) |
| The presence of community members in the health planning teams | 11(2.6) | 28(6.6) | 55(13.0) | 119(28.2) | 209(49.5) |
| Improved health literacy in the society, empowering individuals to make informed | 4(1.0) | 19(4.5) | 18(4.3) | 145(34.4) | 236(55.9) |

**Table 9. Motivations for the use of health research evidence.**

| Variable | SDSA n (%) | DSA n (%) | Neutral n (%) | AG n (%) | SAGR n (%) |
|---|---|---|---|---|---|
| **Automatic motivations** | | | | | |
| Provision of incentives to health planners | 19(4.5) | 21(5.0) | 13(3.1) | 65(15.4) | 304(72.0) |
| Availability of job training | 3(0.7) | 4(1.0) | 16(3.8) | 90(21.3) | 309(73.2) |
| Availability of Short-term courses | 5(1.2) | 13(3.1) | 21(5.0) | 75(17.8) | 308(73.0) |
| Availability of Long-term course | 18(4.3) | 24(5.7) | 24(5.7) | 105(24.9) | 251(59.5) |
| **Reflective motivations** | | | | | |
| Presence of interaction between policymakers, implementers, researchers, and academicians | 3(0.7) | 17(4.0) | 21(5.0) | 111(26.3) | 270(64.0) |
| Presence of active stakeholder engagement in health planning | 6(1.4) | 16(3.8) | 33(7.8) | 116(27.5) | 251(59.5) |
| Continuous quality improvement in the healthcare system | 1(0.2) | 10(2.4) | 12(2.8) | 131(31.0) | 268(63.5) |
| Availability of transparency and accountability mechanisms | 5(1.2) | 12(2.8) | 27(6.4) | 98(23.2) | 280(66.4) |

**Table 10. Looking for evidence that planning teams use health research evidence in their plans.**

| Variable | No | Yes | Not applicable |
|---|---|---|---|
| | n (%) | n (%) | n (%) |
| Do you have a current fiscal year health plan? | 1(1.1) | 87(97.8) | 1(1.1) |
| Do you have a planning guideline? | 1(1.12) | 88(98.9) | 0(0.0) |
| Presence of CCHP | 12(13.5) | 63(70.8) | 14(15.7) |
| HF Planning Guide | 13(14.6) | 74(83.2) | 2(2.3) |
| RHMT guide | 26(29.2) | 20(22.5) | 43(48.3) |
| CHOP | 30(33.7) | 24(27.0) | 35(39.3) |
| Presence of situational analysis in the plan | 8(9.0) | 80(89.9) | 1(1.1) |
| Presence of invitation letters to attend various scientific conferences | 22(24.7) | 67(75.3) | 0(0.0) |
| Are the well-cited plans | 50(56.2) | 39(43.8) | 0(0.0) |
| Is there a budget for research available | 69(77.5) | 20(22.5) | 0(0.0) |
| Is there a research-allocated budget in their health plan? | 68(76.4) | 21(23.6) | 0(0.0) |
| Is the research coordinator among the planning team members who prepared the plan | 64(71.9) | 25(28.1) | 0(0.0) |

having a planning guideline. Most plans include key health frameworks and guides, with 70.8% having the CCHP, 83.2% using the HF Planning Guide, and 89.9% incorporating situational analysis. Scientific engagement is evidenced by 75.3% of teams receiving invitation letters for conferences. However, challenges remain, as only 22.5% have a budget for research, and only 23.6% allocate a budget specifically for research within the health plan. Additionally, just 28.1% include a research coordinator in the planning team, and 43.8% of the plans are well-cited. This highlights a strong foundation in research usage but also gaps in financial and personnel support for research-driven planning.

## Discussion

This study presents the first comprehensive analysis of the use of health research evidence in Tanzania, guided by the COM-B model, which explores Capability, Opportunity, and Motivation as key determinants of behavior. The findings reveal that these three factors significantly influence the extent to which planners utilize health research evidence in decision-making processes.

Planners with higher education levels, more than ten years of experience in health planning, and prior knowledge of research were more likely to integrate evidence into planning. Several determinants were identified as enhancing opportunity, including the availability of organizational support infrastructure such as computers and internet access, the presence of research experts and coordinators, and the role of academic institutions that facilitate the dissemination of research findings.

While a large proportion of participants reported frequent use of routine data and guidelines, a finding consistent with similar studies [23], this trend may indicate an institutional preference for readily available data over more novel research evidence [24,25]. For instance, although 66.2% of respondents reported using health research evidence, this usage may reflect the structured nature of the planning process rather than a deliberate and critical integration of new research findings.

Despite widespread recognition of the importance of using evidence, actual utilization appears to be hindered by limited dissemination practices and a lack of localized, actionable data challenges that have also been observed in other low- and middle-income countries (LMICs). The study also highlighted disparities in capability. Respondents with certificate or diploma-level qualifications reported limited training and research skills, which constrained their ability to use evidence effectively. This finding aligns with literature from similar contexts, where insufficient capability is a major barrier to evidence-informed planning [23–25].

Opportunity-related factors, such as internet access, the presence of research coordinators, and collaboration with academic institutions, played a critical role in enabling evidence use. However, these enablers were not uniformly available across regions, reflecting infrastructural inequities. Such disparities may help explain the regional and facility-level differences observed in evidence use, underscoring the influence of local context and resource allocation [26,27].

Motivational factors, especially automatic motivations such as incentives and professional peer interactions, also emerged as strong influences. Planners who perceived evidence use as aligning with their professional goals were more likely to adopt it. Nevertheless, intrinsic motivation alone may not suffice in the absence of a national roadmap for evidence use [26,27]. These findings are consistent with the COM-B model's assertion that sustainable behavior change requires concurrent improvements in capability, opportunity, and motivation [10].

This study has several strengths. It offers a nuanced and context-specific understanding of the determinants of health research evidence use through a quantitative lens. However, it is important to note some limitations. First, the cross-sectional design precludes causal inference. Second, the study does not capture trends over time. Finally, although the sample was diverse, the findings may not fully reflect variations across all Tanzanian regions or levels of the health system. Future research could benefit from longitudinal designs and the inclusion of more robust measures of actual evidence use.

## Conclusion

Enhancing the use of health research evidence in Tanzania requires a multi-pronged approach that targets capacity building, improved access to resources (opportunities), and the strengthening of motivational frameworks. These findings support previous recommendations in the literature and underscore the need for policies that systematically integrate health research evidence into planning processes. Developing clear frameworks for evidence use and increasing interdisciplinary engagement will be essential in moving toward a more evidence-informed health system.

## Supporting information

**S1 Appendix. PeerJ.** https://doi.org/10.7717/peerj.3077/supp-1.
(DOCX)

## Acknowledgments

We would like to acknowledge the contribution of the University of Dodoma faculty members from the School of Nursing and Public Health for their input in the process of developing this paper. We would like to express our sincere thanks to the study facilities, the RHMTs, and the CHMTs for their support. We are grateful to the research assistants (Edmund

Bunyaga, Theresia Ngungulu, Jestina Nyondo, Devina Mafuta, Micky Masanyaji, and Ally Kinyaga) for their assistance during data collection.

## Author contributions

**Conceptualization:** Pius Kagoma.

**Data curation:** Pius Kagoma.

**Formal analysis:** Pius Kagoma.

**Investigation:** Pius Kagoma.

**Project administration:** Pius Kagoma.

**Software:** Pius Kagoma.

**Supervision:** Richard Mongi, Albino Kalolo.

**Validation:** Pius Kagoma, Albino Kalolo.

**Writing – original draft:** Pius Kagoma.

**Writing – review & editing:** Richard Mongi, Albino Kalolo.

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
