## [Decision Letter · Decision Letter 0]

17 Feb 2025

PONE-D-24-56677Analyzing the determinants for using health research evidence in health planning in Tanzania: a cross-sectional study.PLOS ONE

Dear Dr. Kagoma,

Thank you for submitting your manuscript to PLOS ONE. After careful consideration, we feel that it has merit but does not fully meet PLOS ONE’s publication criteria as it currently stands. Therefore, we invite you to submit a revised version of the manuscript that addresses the points raised during the review process.

- In the methodology section, you have narrated the scoring tools (likert scales and questions), but no source was cited. Please insert the citation for the source(s) in the text with their corresponding references in the reference list.

- Please read critically the comments raised by the reviewer and address each comment accordingly. 

- One more reviewer may be invited to review your original submission before the final decision is made. Thus, you may be required to address the anticipated comments. 

We look forward to receiving your revised manuscript.

Kind regards,

Festo Casmir Shayo, M.D., M.Med., PhD.

Academic Editor

PLOS ONE

5. We note that Figure 2 in your submission contain [map/satellite] images which may be copyrighted. All PLOS content is published under the Creative Commons Attribution License (CC BY 4.0), which means that the manuscript, images, and Supporting Information files will be freely available online, and any third party is permitted to access, download, copy, distribute, and use these materials in any way, even commercially, with proper attribution. For these reasons, we cannot publish previously copyrighted maps or satellite images created using proprietary data, such as Google software (Google Maps, Street View, and Earth). For more information, see our copyright guidelines: http://journals.plos.org/plosone/s/licenses-and-copyright.

1. You may seek permission from the original copyright holder of Figure 2 to publish the content specifically under the CC BY 4.0 license. 

Reviewers' comments:

Reviewer's Responses to Questions

**Comments to the Author**

1. Is the manuscript technically sound, and do the data support the conclusions?

Reviewer #1: No

Reviewer #2: Yes

2. Has the statistical analysis been performed appropriately and rigorously? 

Reviewer #1: No

Reviewer #2: I Don't Know

3. Have the authors made all data underlying the findings in their manuscript fully available?

Reviewer #1: No

Reviewer #2: Yes

4. Is the manuscript presented in an intelligible fashion and written in standard English?

Reviewer #1: No

Reviewer #2: Yes

5. Review Comments to the Author

Reviewer #1: General comment: The study is not clear. One cannot understand what exactly was being investigated. The topic under investigation was not brought up very clearly. Authors should have explained the use of health research evidence and its purpose in the abstract and introduction clearly.

Specific comments:

1. Methodology-The methodology is not clear. For example, there is no specific subsection to describe the characteristics of the study participants. Also, calculation of the sample size was wrongly done by considering marginal error of 50% which does not exist.

-Definition or measurement of the dependent variable which is use of health research evidence is not clear. Authors stated that four questions were used for assessing the use of health research evidence without stating the source of this approach. This affects the authenticity and also reproducibility of the results from the current study.

Results: Are not precise and lack a clear message to the readers.

Reviewer #2: I have read the manuscript and I found it good and I do not have any significant comments regarding it. Introduction is good, methodology was described very well. Results also were clear. Discussion and conclusion were good as well

6. PLOS authors have the option to publish the peer review history of their article (what does this mean? ). If published, this will include your full peer review and any attached files.

**Do you want your identity to be public for this peer review?** For information about this choice, including consent withdrawal, please see our Privacy Policy .

Reviewer #1: **Yes: ** James Yahaya

Reviewer #2: No

---

## [Author Response · Author response to Decision Letter 0]

14 Mar 2025

P.O Box 1923,

Dodoma.

14th March 2025.

Editor-In-Chief,

PLOS ONE

RE: Submission of Revisions of the Manuscript for Publication

I would like to submit the revisions of the manuscript titled “Analyzing the Determinants for Using Health Research Evidence in Health Planning in Tanzania: A Cross-sectional Study”. This manuscript presents the findings of a quantitative study aimed at understanding the factors influencing the use of health research evidence among health planning teams in Tanzania.

The relevance of this research lies in its potential to inform policy and practice interventions aimed at enhancing the utilization of evidence-based practices in healthcare planning, particularly in low-resource settings including Tanzania.

This manuscript is a research article and a new submission to PLOS ONE. The content of this manuscript has not been published or submitted elsewhere for consideration.

Sincerely,

Dr. Pius Kagoma

Corresponding Author

---

## [Decision Letter · Decision Letter 1]

6 Apr 2025

PONE-D-24-56677R1Analyzing the determinants for using health research evidence in health planning in Tanzania: a cross-sectional study.PLOS ONE

Dear Dr.  Kagoma,

Thank you for submitting your manuscript to PLOS ONE. After careful consideration, we feel that it has merit but does not fully meet PLOS ONE’s publication criteria as it currently stands. Therefore, we invite you to submit a revised version of the manuscript that addresses the points raised during the review process.

 Please submit your revised manuscript by May 21 2025 11:59PM. If you will need more time than this to complete your revisions, please reply to this message or contact the journal office at plosone@plos.org . Please include the following items when submitting your revised manuscript:

A rebuttal letter that responds to each point raised by the academic editor and reviewer(s). You should upload this letter as a separate file labeled 'Response to Reviewers.'A marked-up copy of your manuscript that highlights changes made to the original version. You should upload this as a separate file labeled 'Revised Manuscript with Track Changes.'An unmarked version of your revised paper without tracked changes. You should upload this as a separate file labeled 'Manuscript.'

We look forward to receiving your revised manuscript.

Kind regards,

Festo Casmir Shayo, M.D, M.Med, PhD.

Academic Editor

PLOS ONE

Additional Editor Comments:

**ACADEMIC EDITOR COMMENTS:**

1. Title: Replace Analyzing with “Analysis of.”

2. Keep only the email of the corresponding author.

3. Abstract:

Introduction should be about the use of health research evidence and not about universal health coverage.Your study object is to investigate the determinants of the utilization of health research evidence and not the barriers to the utilization of health research evidence. To align with your study objective, please remove all results and discussion regarding barriers to the use of health research evidence. You can produce another manuscript on the barriers associated with the utilization of health research evidence. Otherwise, rephrase your title to include the “barriers for” ……. For instance, **“Analysis of the determinants and barriers for using health research evidence in health planning in Tanzania.”**

4. Results:

Delete all introductory sections of each results subheading: page 12 lines 285-289; page 16 lines 323-328; and other similar sections throughout the results section. They are not relevant to the research article and are reserved for the submission of the dissertation report.In regression analysis, describe only the results for the adjusted model. The unadjusted model can be traced in the table.Avoid describing the analysis parameters. Describe the intended results right away. For example: “In the adjusted regression model, the study participants from regional referral hospitals were significantly more likely to use health research evidence than those from dispensaries: (aOR = 2.67, CI: [1.89-7.92], p-value = 0.038).”In regression analysis, use “than” instead of “compared to” (used in the chi-square test).In all tables of results, please remove all vertical lines and retain only three horizontal lines: one under the title, one above the column heading, and one between the column headings and the body of the table.I suggest you download and read at least three original research articles from the PLOS One Journal and learn how the results are presented. Your results are presented in the dissertation report format.  

5. Discussion:

Your interpretation of the findings lacks an alternative explanation of the results and study strengths and limitations. Also, maintain objectivity and logical flow, and avoid repeating the results section. Please download and read at least three original research articles from the PLOS One Journal to improve your discussion and conclusion sections.  

5. Conclusion

Keep your conclusion relatively short by summarizing the most striking findings relevant to your study objective(s)

Reviewers' comments:

Reviewer's Responses to Questions

**Comments to the Author**

1. If the authors have adequately addressed your comments raised in a previous round of review and you feel that this manuscript is now acceptable for publication, you may indicate that here to bypass the “Comments to the Author” section, enter your conflict of interest statement in the “Confidential to Editor” section, and submit your "Accept" recommendation.

Reviewer #1: (No Response)

Reviewer #3: All comments have been addressed

2. Is the manuscript technically sound, and do the data support the conclusions?

Reviewer #1: No

Reviewer #3: Yes

3. Has the statistical analysis been performed appropriately and rigorously? 

Reviewer #1: No

Reviewer #3: I Don't Know

4. Have the authors made all data underlying the findings in their manuscript fully available?

Reviewer #1: No

Reviewer #3: Yes

5. Is the manuscript presented in an intelligible fashion and written in standard English?

Reviewer #1: No

Reviewer #3: Yes

6. Review Comments to the Author

Reviewer #1: After evaluating the revised version of the manuscript, in my views, I don't think that the work meets the quality to be published. I think authors have failed to take time and make significant changes. After just checking the abstract, you realized serious errors especially in the results section. Could be that the work was left in the hands of a person who is not competent in writing and articulating research. This has made me to just end up the review process with only abstract.

Reviewer #3: Authors have adequately addressed all of the previous reviewers comments. I myself did not review this article previously but I reviewed the author responses to the previous review.

7. PLOS authors have the option to publish the peer review history of their article (what does this mean? ). If published, this will include your full peer review and any attached files.

**Do you want your identity to be public for this peer review?** For information about this choice, including consent withdrawal, please see our Privacy Policy .

Reviewer #1: **Yes: ** JAMES Joseph YAHAYA

Reviewer #3: No

---

## [Author Response · Author response to Decision Letter 1]

16 Apr 2025

P.O Box 1923,

Dodoma.

16th April 2025.

Editor-In-Chief,

PLOS ONE

RE: Submission of Revision of the Manuscript for Publication

I would like to submit the revision of the manuscript titled “Analysis of the Determinants for Using Health Research Evidence in Health Planning in Tanzania: A Cross-sectional Study”. This manuscript presents the findings of a quantitative study aimed at understanding the factors influencing the use of health research evidence among health planning teams in Tanzania.

The relevance of this research lies in its potential to inform policy and practice interventions aimed at enhancing the utilization of evidence-based practices in healthcare planning, particularly in low-resource settings, including Tanzania.

This manuscript is a research article and a revised submission to PLOS ONE. The content of this manuscript has not been published or submitted elsewhere for consideration.

Sincerely,

Dr. Pius Kagoma

Corresponding Author

---

## [Editor Report · Decision Letter 2]

21 Apr 2025

Analysis of the determinants for using health research evidence in health planning in Tanzania: a cross-sectional study.

PONE-D-24-56677R2

Dear Dr. Kagoma,

We’re pleased to inform you that your manuscript has been judged scientifically suitable for publication and will be formally accepted for publication once it meets all outstanding technical requirements.

Kind regards,

Festo Casmir Shayo, M.D, M.Med, PhD.

Academic Editor

PLOS ONE
---

## [Editor Report · Acceptance letter]

PONE-D-24-56677R2

PLOS ONE

Dear Dr. Kagoma,

I'm pleased to inform you that your manuscript has been deemed suitable for publication in PLOS ONE. Congratulations! Your manuscript is now being handed over to our production team.

Kind regards,

on behalf of

Dr. Festo Casmir Shayo

Academic Editor

PLOS ONE